# Drug supply management at first-level public health facilities: Case of Pyay District, Myanmar

**Thein Hlaing** [1]*, **Tun Win Lat** [2]

1 Township Public Health Department, Zigon Township, Bago Region, Myanmar, 2 Township Public Health Department, Paungde Township, Bago Region, Myanmar

* theinhlaing231@gmail.com

**Data Availability Statement:** All collected data and synthesized information were displayed in the forms of words and tables in the finding portion and more information can be enquired from the Institutional Review Board, Naypyitaw, Myanmar,

## Abstract

First-level public health facilities (PHFs) serve as primary providers of essential medicines, necessitating critical attention to drug availability and quality assurance. This study aimed to examine the status of functional areas within the drug supply chain management framework and assess the overall capability maturity at first-level PHFs. The cross-sectional study was conducted among 183 drug store sites from six townships of Pyay District. Only situational analysis was exercised to determine the existing situations. The overall capability maturity was determined according to the definitions of levels of the Capability Maturity Module Tool. 58.47% lacked formal drug supply management training, with 23.5% not undergoing performance reviews. Drug forecasting predominantly relied on a pen-paper system (91.6%) and factors like patient load (87.39%), drug consumption (85.71%), and disease prevalence (64.71%). Store site analysis revealed that 65.03% exhibited marginal capability, lacking standardized drugstores and employing unstandardized procedures. Storage practices varied, with 48.69% storing drugs conveniently and others categorizing them by drug type (32.79%) or using the first-expired-first-out system (40.98%). Approximately 42.69% reported having expired drugs. Concerning transportation costs, 37.16% incurred expenses exceeding 20,000 Kyats per time, with management staff often covering the costs. Waste management methods included burial pits (49.18%), incineration (62.84%), and sharp pits (55.19%). A majority (78.14%) used safety boxes, and 57.38% implemented a color-coded system for waste bins. The logistics management information system was entirely paper-based (100%). On average, assessments of drug quality conditions and physical damages scored 46.51% and 48.20%, respectively. The overall supply chain maturity at first-level public health facilities is at a marginal capability level (36.35%). While some basic drug supply chain management procedures were in place, they were not consistently followed, and many systems remain manual. The findings underscored significant inconsistencies in the management functions of supplied drugs, with poor adherence to Standard Operating Procedure guidelines.

irb1@mohs.gov.mm. The authors have attached two files; Inform Consent Form and Human Subject Research Checklist regarding ethical consideration. Besides, the detailed information about ethical consideration has already mentioned under the sub-title "Ethical Consideration" in the section of "Material and method".

**Funding:** The required funds for conducting this research were granted by Ministry of Health and Sports (MOHS) Research Grant, Myanmar, (2022-2023). The grant number is 089/2022/MOHS. Date of receiving grant is 2 December 2022. The research grant is only for the study. The funder had no role in the study design, data collection and analysis, decision to publish, or preparation of manuscript.

**Competing interests:** The authors have declared that no competing interests exist.

## Introduction

Primary healthcare and the provision of essential medicines constitute the primary functions of first-level public health facilities. Operating under the auspices of the Myanmar Ministry of Health, the National Supply Chain Management System (NSCMS) has transitioned to a "pull-based supply chain," driven by demand, as opposed to the erstwhile "push-based supply chain" characterized by central control over drug needs. Notably, NSCMS increased the public health spending share from 11.4% in 2009 to 23.9% in 2012, aiming to enhance the availability and relevance of crucial essential drugs across medical and public health sectors [1]. These strategic actions aim at bolstering the drug supply chain system in Myanmar. NSCMS extends technical assistance to three national programs (acquired immunodeficiency syndrome (AIDS), Tuberculosis, and Malaria) and other sub-recipients. Additionally, it oversees warehouses and storage facilities at State and Regional procurement and supply management sections. The NSCMS management's responsibilities encompass training procurement and supply system management staff in medicinal forecasting, ordering, procurement, receiving, storing, distributing processes, and the logistics management information system. Health spending is allocated to the State, Regional, District, and Township departments for their individual procurement and supply management. Rural Health Centers (RHC) and Sub-Rural Health Centers (Sub-RHC) retrieve key essential medicines from their respective township health departments or hospitals, with their procurement handled externally. However, they are responsible for forecasting drug needs, store and inventory management, transportation and distribution, waste management, and information system logistics management. Formal training and refresher courses are provided to equip the public health professional with the essential skills [2].

In Myanmar, it is important to collaborate with basic public health professionals to manage drug supply at first-level public health facilities. Their understanding of correct supply chain processes at their respective levels is paramount. All facility staff utilize essential drugs to implement various components of primary health care, including accessible and affordable healthcare services, treatment of common ailments, communicable and non-communicable disease controls, basic medical care for mental health, maternal and child health, nutritional management, pre-referral treatments, local endemic disease control, and emergency management [3]. Public health facilities, including RHC and Sub-RHC, play a pivotal role as providers of essential medicines, where drug availability and quality assurance are imperative. The distribution of substandard drugs can lead to unsuccessful treatments, drug resistance development, and adverse impacts on individual health [4]. Effective budgeting and procurement necessitate standardized and robust methods for forecasting drug needs at every provider site. The budgeting exercise, forecasting, and supply planning of the NSCMS rely on evidence-based estimations from provider sites under the "pull" system. Accurate forecasting supports optimal allocation, and budget control, and prevents wastage of medicine [5, 6].

Inventory management emerges as another critical factor for effective drug supply management at all levels. It requires a regular and organized approach, aligning with recommended guidelines, to ensure the proper storage of drugs and prevent issues such as stockouts and expiries [7]. This comprehensive approach ensures the sustainability and efficiency of drug supply systems, safeguarding patient outcomes and minimizing wastage. As per Tolliver and Bartram's report, numerous drug stores in Myanmar face challenges associated with age and overcrowding. These conditions impede store management staff from reaching their optimal potential to adhere to best-standard guidelines for drug store management [2]. The report underscores the pivotal role of effective and efficient medical supply maintenance in enhancing the utilization rate of outpatient departments (OPD). Jha and Mahatme et al., similarly

emphasize the significance of store management tasks in balancing existing budgets and addressing drug necessities. This includes prioritizing purchases and distributions, ensuring adequate stock, preventing pilferage, and strategically reallocating nearly expired drugs [8, 9]. The findings from Tolliver and Bartram's report also shed light on the absence of a defined waste management procedure for public health products and the lack of clear instructions on managing public healthcare waste at any level [2]. Addressing these issues is crucial for optimizing healthcare resources and ensuring the effective functioning of medical facilities.

According to the report by Tolliver and Bartram, numerous public health sector supply chains in Myanmar lack standardized drug stores. These stores, ideally situated in isolated, shaded, accessible, and secure locations with sturdy structures to prevent environmental damage and pest infestation, are deemed crucial for proper drug storage [2]. The absence of such infrastructure raises concerns about maintaining drug quality and the availability of safety equipment [10]. The medicines distribution system of Central Medical Store Depot (CMSD) and vendors mainly focus on the State/Regional and Township health departments and major hospitals, not including RHC and Sub-RHC. The report highlights high stockout rates and the presence of expired drugs within many public health sector supply chains. Additionally, it identifies inadequacies in the healthcare waste management system. This underscores the importance of evaluating the drug supply management processes at first-level public health facilities, particularly how RHC and Sub-RHC maintain drug quality without dedicated drug stores. Key considerations include storage practices, transportation modes from township departments to RHC and Sub-RHC, cost resolution, stockouts management, and management of nearly expired and expired drugs. The assessment also extends to the accuracy of inventory management and drug forecasting procedures at RHC and Sub-RHC. Furthermore, scrutiny of the applied methods and procedures for healthcare waste management is imperative to ensure hygiene and safety for communities and healthcare providers. The potential challenges and obstacles in the entire public health sector supply chain process needs to be systematically investigated and analyzed. Notably, there is a lack of published Myanmar studies on drug supply management at first-level public health facilities, making this research output a critical baseline. The study aims to evaluate the obstacles and challenges in drug supply management at first-level public health facilities, with specific objectives including assessing supply chain management training, drug forecasting planning, drugstore and inventory management, transportation and delivery issues, waste management system, logistics management information system, quality control procedures, and overall capability maturity of the drug supply chain.

## Material and method

### Ethics statement

The research adhered to strict ethical standards, receiving approval from the Institutional Review Board (Nay Pyi Taw), Ministry of Health, Myanmar. The study, approved under IRB number 2023–06, emphasized ethical practices throughout its execution. Participation was voluntary, with formal consent obtained through detailed forms translated into Myanmar and explained verbally when necessary. Anonymity and confidentiality were maintained, with data securely stored for five years and restricted access to authorized personnel. Results were presented accurately, and precautions were taken to avoid harm during data collection.

### Study design and scope

This research employed a cross-sectional and descriptive approach to examine the drug supply chain at first-level health facilities (RHC and Sub-RHC) at a specific point-in time. The

investigation focused solely on the first-level public health drug supply chain, excluding any assessment of the supply chain levels of the township public health department, township hospital, and station hospital. The study comprehensively evaluated all functional areas of the current medicinal supply chain, including CMSD, Nutrition, Tuberculosis (TB), Human Immunodeficiency Virus (HIV), Malaria, Epilepsy, Leprosy, and Non-communicable diseases (NCD), distributed by the township public health department. However, it did not assess the vaccine-related supply chain and other supplies provided by the community and local donors.

## Study settings and population

The study targeted all 43 RHCs, 6 Maternal and Child Health Centers (MCHs), and 134 Sub-RHCs, totalling 183 public health facilities within Pyay District, Bago Region. Face-to-face interviews were conducted with the health personnel managing the drug stores (a total of 183) from each RHC and Sub-RHC. Pyay District was chosen as the study area due to the active functionality of all public health facilities, the accelerated utilization of supplied drugs in OPD clinics, NCD clinics, and special clinics for retired persons, and the availability of representative and required data in this district.

## Data collection techniques and sources of information

Primary data were gathered through face-to-face interviews, self-administration, and observations. Background characteristics of public health facilities and drug supply management staff, training information, self-perceived capacity for drug forecasting and planning, infrastructures and safety equipment in the drugstore, and conditions of stockouts of supplied drugs were collected through self-administration. Observations covered the conditions of the place where the supplied drugs are stored and all relevant documents of drug supply management. Face-to-face interviews were conducted to gather information on several functional areas of the drug supply chain. These areas included forecasting drug requirements, storage procedures, ordering, receiving and dispensing supplied drugs, transportation and delivery issues, waste management, the logistics management information system, and quality control procedures.

## Preparation of data collection tools

The semi-structured interview questionnaire was developed based on various sources, including training materials and checklists from the World Health Organization (WHO)/ Child Health Development (CHD) and Basic Support for Institutionalizing Child Survival (BASICS) [11], the capability maturity model tool by Tolliver and Bartram (2014) [2], the USAID Global Health Supply Chain Program's capability maturity module questionnaire (2019) [12], drug store guidelines from MOH Myanmar (2016) [13], and checklists from Basic Support for Institutionalizing Child Survival [11]. The questionnaire consisted of open-ended questions, multiple-choice questions and "Yes or No" questions. It covered background information, training, drug forecasting and planning, drug storage and inventory management, transportation and delivery issues, waste management, the logistics supply management information system, and quality control procedures. Various checklists, such as those for physical conditions of the drug store, drug storage procedures, bin cards, drug requisition forms and ordering drug supplies, receiving drug supplies, and dispensing procedures, were applied for structuring the research questionnaire. The questionnaire underwent testing for face and content validity, computation of Cronbach alpha values, and subsequent revisions to ensure reliability.

## Assessment of capability maturity of drug supply chain

This study aimed to evaluate the capability maturity of the drug supply chain at first-level health facilities, utilizing a modified capability maturity model (CMM) proposed by Tolliver and Bartram [2]. The CMM, adapted for measuring five maturity levels—minimal, marginal, qualified, advanced practice, and best practice—was applied to assess various functional areas within the drug supply chain (Tables 1 and 2).

To determine the capability maturity of each functional area, the researcher rated the areas on a scale of 1–20%, 21–40%, 41–60%, 61–80%, and 81–100% upon completion of each maturity level.

## Training of data collectors

A proficient data collection team, consisting of ten members comprising retired public health supervisors, midwives, and lady health visitors with fundamental medical knowledge, was assembled. Two members conducted face-to-face interviews, two served as observers, and another guided participant in self-reporting. The team underwent comprehensive training on all data collection instruments and a concise training guide. A pilot study was conducted in selected RHC and Sub-RHC within Nattalin Township, Bago Region, to assess the feasibility of data collection instruments, evaluate the data collectors' comprehension of methods and procedures, and estimate the time required for data collection.

## Data collection

Before data collection, the research objectives and contents were communicated to all regional, district, and township public health authorities, with the researcher advocating for stakeholder participation within Pyay District. The data collection plan aligned with the numbers and locations of primary public health facilities in the chosen township, with five to six rural health facilities visited a day. The data collection was started on 31st March 2023 and ended on 9th May 2023. The entire data collection process spanned approximately 40 working days, strictly adhering to the current COVID-19 prevention guidelines issued by central and local health authorities. Supervisors (principal researcher and co-researcher) played a crucial role in maintaining a positive relationship between data collectors and participants, addressing unexpected challenges, ensuring adherence to field data collection protocols, and overseeing the secure storage of research questionnaires and checklists. Supervision was conducted daily, encompassing in-person and tele-supervision.

## Data management and analysis

The collected data underwent coding and entry into a Statistical Package for the Social Sciences (SPSS) spreadsheet. SPSS software was then employed for data cleaning, correction, and transformation into the desired format for subsequent analysis. This study employed situational analysis to identify obstacles and challenges in the implementation of the drug supply chain at first-level health facilities. Frequencies and proportions were computed to list and rank different types of obstacles and challenges in each functional area of drug supply management. Strengths, weaknesses, and risk factors for functional development were determined by setting 50th percentiles based on the average scores of each functional area. Additionally, the overall maturity of the first-level health facility supply chain was computed using the average score of each functional area, classified into five levels (1–20%, 21–40%, 41–60%, 61–80%, and 81–100%), and interpreted according to the definitions of the five levels of the CMM tool.

**Table 1. Assessment of capability maturity of drug supply chain.**

| Functional Areas | Level | Explanation |
|---|---|---|
| **Capacity-building** | 1 (Minimal Capability) | Capacity-building primarily relied on on-the-job training, learning from experiences, and informal training such as Continuous Medical Education (CME). It lacked practical training guidelines. |
| | 2 (Marginal Capability) | Formal training and on-the-job learning were present, but the practical application of training guidelines and instructions was inconsistent. |
| | 3 (Qualified Capability) | Capacity-building included formal training, refresher training, and online learning with practical application of guidelines, but lacked consistent close guidance and supportive supervision. |
| | 4 (Advanced Practice Capability) | Capacity-building encompassed formal training, refresher training, and online learning with correct and consistent application of guidelines, albeit with irregular close guidance and supervision. |
| | 5 (Best Practice Capability) | Capacity-building, including formal training and the use of LMIS-integrated software tools, was consistently applied with regular close guidance and supportive supervision. |
| **Drug forecasting planning** | 1 (Minimal Capability) | Drug forecasting planning was irregular and lacked Standard Operation Procedures (SOPs), relying on convenient procedures. |
| | 2 (Marginal Capability) | SOPs and skilled staff were present, but planning remained irregular and relied on convenient procedures. |
| | 3 (Qualified Capability) | SOPs and skilled staff were utilized for regular drug forecasting planning based on various factors such as service data, demographic data, drug consumption/issues data, disease prevalence, previous forecasting data, and budget. |
| | 4 (Advanced Practice Capability) | SOPs and skilled staff guided regular drug forecasting planning based on comprehensive factors. |
| | 5 (Best Practice Capability) | SOPs and skilled staff integrated with a Logistic Management Information System (LMIS) for advanced and efficient drug forecasting planning. |
| **Drugstore** | 1 (Minimal Capability) | The drugstore existed but lacked standardization, adequate size, and SOPs/guidelines. |
| | 2 (Marginal Capability) | The drugstore had SOPs/guidelines, but supplied drugs were stored without consistent adherence to SOPs/guidelines. |
| | 3 (Qualified Capability) | A standardized drugstore with SOPs/guidelines was present, but some drugs were not stored following standardized procedures. |
| | 4 (Advanced Practice Capability) | A standardized drugstore with SOPs/guidelines and electricity was maintained, ensuring all supplied drugs were stored with standardized procedures. |
| | 5 (Best Practice Capability) | A standardized drugstore with SOPs/guidelines, electricity, and engines or solar systems was in place, guaranteeing all supplied drugs were stored with standardized procedures. |
| **Inventory management** | 1 (Minimal Capability) | Utilizing a pull system but lacking SOPs, receiving and dispensing supplied drugs without standardized procedures. |
| | 2 (Marginal Capability) | Implementing a pull system and SOPs, receiving and dispensing supplied drugs with standardized procedures. |
| | 3 (Qualified Capability) | Utilizing a pull system, SOPs, and an owned-computer system for receiving dispensing, and inventory management. |
| | 4 (Advanced Practice Capability) | Utilizing a pull system, SOPs, and a government-owned computer system for receiving dispensing, and inventory management. |
| | 5 (Best Practice Capability) | Employing a pull system, SOPs, and a government-owned computer system linked with upper levels using network software for advanced inventory tracking and management. |

(*Continued*)

**Table 1.** (Continued)

| Functional Areas | Level | Explanation |
|---|---|---|
| **Waste management** | 1 (Minimal Capability) | Informal waste management practices by health staff with no defined waste handler. |
| | 2 (Marginal Capability) | SOPs/guidelines and a designated waste handler were present, but the waste handler lacked training. |
| | 3 (Qualified Capability) | SOPs/guidelines, a trained waste handler, and government budget support for waste management infrastructure were available. |
| | 4 (Advanced Practice Capability) | SOPs, a trained waste handler, government budget support, and functional waste management infrastructures were in place. |
| | 5 (Best Practice Capability) | Availability of private sector waste management services directly to the health facility. |

## Research period

The recruitment period for this study commenced on March 31, 2023, and concluded on December 31, 2023. This timeframe encapsulated an extended window during which participants were actively sought and enlisted for the research endeavour. The nearly two-year span allowed for a comprehensive approach to participant recruitment, ensuring a diverse and representative sample for the study.

## Results

### Background characteristics of public health facilities and drug supply management staff

The survey and interviews encompassed 183 first-level public health facilities and an equivalent number of drug supply management staff. Among these, 6 (3.28%) were MCH, 43 (23.49%) were RHC, and 134 (73.22%) were Sub-RHC. The majority (85.79%) possessed a main health facility building, but 56.05% required repairs. Notably, 35.03% were over 10 years old, with 50.32% repurposed as staff houses. Among the drug supply management staff, 91.80% were female, with graduates (82.51%) and midwives (MW) (82.51%) representing the predominant educational and professional backgrounds. Concerning public sector service length, 45.36% had less than or equal to 10 years, while 54.64% exceeded 10 years. The comprehensive background characteristics of the study sample are outlined in Table 3.

### Drug supply management training

Regarding training in drug supply management, 41.53% reported having no training, 33.88% had completed one course, and 7.65% had attended multiple courses. Of those trained, 34.21% received training before 2017, with 75% participating in refresher training and 64.47% in typical training. Regarding comprehension, 36.84% understood a quarter, 28.95% half, 25.00% about two-thirds, and 9.21% more than three-quarters of the latest training. Additionally, 42.08% lacked training guidelines. Performance evaluations for drug management staff were conducted quarterly or more often (30.05%), bi-annually (32.24%), annually (11.48%), less frequently than annually (2.73%), and 23.5% were never reviewed. In the past year, 17.49% received supportive supervision, with 84.38% obtaining feedback and corrective actions. Further details are available in Table 4.

**Table 2. Interpretations of supply chain maturity levels of CMM tool.**

| Sr. No. | Resulted in Maturity Score | Maturity Level | Interpretation |
|---|---|---|---|
| 1 | 0%–20% | Minimal Level | Drug supply chain management is informally processed without following many guidelines, instructions, and systems. |
| 2 | 21%–40% | Marginal Level | Basic drug supply chain management procedures are formally in place, but they are inconsistent, and most systems are manual. |
| 3 | 41%–60% | Qualified Level | The drug supply chain management is well-defined and documented, and some supply chain technology is used. |
| 4 | 61%–80% | Advanced Practice Level | The drug supply chain management is well-defined and documented and supply chain technology is internally integrated. |
| 5 | 81%–100% | Best Practice Level | The formal processes of drug supply chain management are continuously practiced and improved, with supply chain technology fully integrated. |

**Table 3. Background characteristics of public health facilities and drug supply management staff.**

| Background characteristics of public health facilities (n = 183) | | | Background characteristics of drug supply management staff (n = 183) | | |
|---|---|---|---|---|---|
| Characteristics | Frequency | Percentage | Characteristics | Frequency | Percentage |
| Township (n = 183) | | | Township (n = 183) | | |
| Pyay | 28 | 15.30 | Pyay | 28 | 15.30 |
| Paukkhaung | 40 | 21.86 | Paukkhaung | 40 | 21.86 |
| Pandaung | 27 | 14.75 | Pandaung | 27 | 14.75 |
| Paungde | 30 | 16.39 | Paungde | 30 | 16.39 |
| Shwedaung | 28 | 15.30 | Shwedaung | 28 | 15.30 |
| Thaegon | 30 | 16.39 | Thaegon | 30 | 16.39 |
| Type of Public Health Facility (n = 183) | | | Age of respondent (Complete years) (n = 183) | | |
| RHC + MCH | 49 | 26.78 | < = 40 Years | 112 | 61.20 |
| Sub-RHC | 134 | 73.22 | > 40 Years | 71 | 38.80 |
| Main Building of Public Health Facility (n = 183) | | | Gender (n = 183) | | |
| Present | 157 | 85.79 | Male | 15 | 8.20 |
| Absent | 26 | 14.21 | Female | 168 | 91.80 |
| Current condition of the Public Health Facility (n = 157) | | | Highest education (n = 183) | | |
| No need to repair | 69 | 43.95 | High school passed | 25 | 13.66 |
| Need to repair | 88 | 56.05 | Diploma | 7 | 3.83 |
| Duration of the current building of the Public Health Facility (n = 157) | | | Graduate | 151 | 82.51 |
| | | | Current title (n = 183) | | |
| < = 10 Years | 102 | 64.97 | Public Health Supervisor-1 (PHS-1) | 1 | 0.55 |
| > 10 Years | 55 | 35.03 | Public Health Supervisor-2 (PHS-2) | 4 | 2.19 |
| Ownership of the Public Health Facility (n = 157) | | | MW | 151 | 82.51 |
| Government | 138 | 87.90 | Lady Health Visitor (LHV) | 13 | 7.10 |
| Public | 19 | 12.10 | Health Assistant (HA) | 14 | 7.65 |
| Use of Public Health Facility as Staff House (n = 157) | | | Length of service in the public health sector (n = 183) | | |
| Yes | 79 | 50.32 | < = 10 Years | 83 | 45.36 |
| No | 78 | 49.68 | > 10 Years | 100 | 54.64 |
| Availability of Staff House (n = 183) | | | Length of service at current health facility (n = 183) | | |
| Present | 56 | 30.60 | < = 5 Years | 64 | 34.97 |
| Absent | 127 | 69.40 | > 5 Years | 119 | 65.03 |

**Table 4. Drug supply management training (n = 183).**

| Descriptions | Frequency | Percentage |
|---|---|---|
| Have you received formal drug supply management training? (n = 183) | | |
| Yes | 76 | 41.53 |
| No | 107 | 58.47 |
| Number of formal drug supply management training received during current position (n = 183) | | |
| 0 time | 107 | 58.47 |
| 1 time | 62 | 33.88 |
| > 1 time | 14 | 7.65 |
| Last training (n = 76) | | |
| Before 2017 | 26 | 34.21 |
| After 2017 | 50 | 65.79 |
| Types of formal drug supply management training* (n = 76) | | |
| Refresher training | 57 | 75.00 |
| Typical training | 49 | 64.47 |
| Types of informal drug supply management training* (n = 107) | | |
| On-the-job-training | 17 | 15.89 |
| Continuous medical education | 86 | 80.37 |
| Training guides and materials | 11 | 10.28 |
| Standard operation procedures | 5 | 4.67 |
| Self-learning | 53 | 49.53 |
| Subjects of last drug supply management training* (n = 76) | | |
| Drug forecasting | 51 | 67.11 |
| Drug requisition form and ordering drug supplies | 48 | 63.16 |
| Drug storage procedures | 43 | 56.58 |
| Receiving drug supplies | 38 | 50.00 |
| Bin cards | 33 | 43.43 |
| Medicine Quality Assurance | 32 | 42.11 |
| Treatment guidelines | 32 | 42.11 |
| Physical conditions of drug store | 29 | 38.16 |
| Waste management system of drug supply chain | 29 | 38.16 |
| Dispensing procedures | 27 | 35.53 |
| LIMS (Logistic Information Management System) | 19 | 25.00 |
| Application of fire extinguishers | 13 | 17.11 |
| Self-perceived understandability of your last drug supply management training* (n = 76) | | |
| 25% | 28 | 36.84 |
| 26–50% | 22 | 28.95 |
| 51–75% | 19 | 25.00 |
| 76–100% | 7 | 9.21 |
| Do you have the drug supply management guidelines? (n = 183) | | |
| Yes | 106 | 57.92 |
| No | 77 | 42.08 |
| Self-perceived understandability of the drug supply management guidelines (n = 106) | | |
| 25% | 51 | 48.11 |
| 26–50% | 14 | 13.21 |
| 51–75% | 31 | 29.25 |
| 76–100% | 10 | 9.43 |
| How often is the performance of supply management staff reviewed? (n = 183) | | |
| Quarterly or more often | 55 | 30.05 |

(*Continued*)

**Table 4.** (Continued)

| Descriptions | Frequency | Percentage |
|---|---|---|
| Bi-annually | 59 | 32.24 |
| Annually | 21 | 11.48 |
| Less frequently than annually | 5 | 2.73 |
| Never | 43 | 23.50 |
| Has the supply management staff received supportive supervision within the last year? (n = 183) | | |
| Yes | 32 | 17.49 |
| No | 151 | 82.51 |
| Do supply chain staff receive feedback after supportive supervision? (n = 32) | | |
| Yes | 27 | 84.38 |
| No | 5 | 15.63 |
| Are corrective actions of the supply management staff taken following supervision visits? (n = 32) | | |
| Yes | 27 | 84.38 |
| No | 5 | 15.63 |
| Are guidelines/checklists for supply chain supervision available? (n = 32) | | |
| Yes | 25 | 78.13 |
| No | 7 | 21.88 |

* Multiple responses

## Strengths, weakness and risks (SWR) analysis of capacity building

In the SWR analysis aimed at assessing the strengths, weaknesses, and risks in the current capacity-building scenario, this study focused on various training variables, including years, frequency, status, types, subjects, and the presence of training guidelines. Each participant received a score out of 21, categorizing those scoring 10.5 (50%) and above (17, 0.29%) as having good capacity building, while the majority (166, 90.71%) scoring below 10.5 (50%) were deemed to have poor capacity building. Notably, 17.49% of participants enhanced their capacities through supportive supervision, while 73.77% did so through performance reviews. The self-perceived understandability of drug supply management procedures was identified as a risk, with 161 participants (87.98%) falling into the risk group due to self-perceived understandability below 50%. In the SWR analysis, guidelines and performance reviews emerged as strengths, while training and supportive supervision were identified as weaknesses, and self-perceived understandability was pinpointed as a risk.

## Forecasting the drug requirements

Among the 183 first-level health facilities surveyed, 34.97% did not forecast drug requirements in the past year, and 63.39% lacked SOPs for forecasting. About 47.54% perceived their capacity for drug forecasting to be only 25%. Most facilities based their forecasts on service data (87.39%), drug consumption/issue data (85.71%), population data (66.39%), disease prevalence (64.71%), and previous forecasting data (33.61%). However, 26.89% used convenient forecasting methods. Over half (54.1%) regularly monitored drug consumption, 31.69% did so occasionally, and 14.2% did not monitor at all. Regarding report submissions, 91.26% submitted their LMIS data to upper levels, with 73.65% doing so monthly, 14.49% half-yearly, and 2.99% bi-monthly. Additional details are in Table 5.

**Table 5. Forecasting the drug requirements.**

| Descriptions | Frequency | Percentage |
|---|---|---|
| **Do you forecast the drug requirements? (n = 183)** | | |
| Yes | 119 | 65.03 |
| No | 64 | 34.97 |
| **SOPs for drug forecasting the drug requirements (n = 183)** | | |
| Yes | 67 | 36.61 |
| No | 116 | 63.39 |
| **Self-perceived capacity for drug forecasting and planning (n = 183)** | | |
| 25% | 87 | 47.54 |
| 26–50% | 35 | 19.13 |
| 51–75% | 51 | 27.87 |
| 76–100% | 10 | 5.46 |
| **Forecasting the drug requirements based on* (n = 119)** | | |
| Service data (Patient load) | 104 | 87.39 |
| Drug consumption/issues data | 102 | 85.71 |
| Demographic data (Population) | 79 | 66.39 |
| Disease prevalence | 77 | 64.71 |
| Previous forecasting planning data | 40 | 33.61 |
| Convenient procedure | 32 | 26.89 |
| Budget | 7 | 5.88 |
| **Develop drug forecasting (n = 119)** | | |
| Every two months | 16 | 13.45 |
| Quarterly | 28 | 23.53 |
| Half-yearly | 4 | 3.36 |
| Annually | 7 | 5.88 |
| Convenient procedure | 64 | 53.78 |
| **Practice for forecasting the drug requirements is to use (n = 119)** | | |
| Software tool integrated with LMIS | 10 | 8.40 |
| Pen and paper | 109 | 91.60 |
| **Do you monitor drug consumption? (n = 183)** | | |
| Yes (Regular) | 99 | 54.10 |
| Yes (Sometime) | 58 | 31.69 |
| No | 26 | 14.21 |
| **Do you submit LMIS reports to upper levels? (n = 183)** | | |
| Yes | 167 | 91.26 |
| No | 16 | 8.74 |
| **Submission of LMIS reports (n = 167)** | | |
| Monthly | 123 | 73.65 |
| Bi-monthly | 5 | 2.99 |
| Quarterly | 7 | 4.19 |
| Half-yearly | 25 | 14.97 |
| Yearly | 7 | 4.19 |
| **LMIS reporting frequency last year (n = 167)** | | |
| < = 4 times | 38 | 22.75 |
| > 4 times | 129 | 77.25 |

* Multiple responses

## SWR analysis of drug forecasting

The analysis focused on variables such as SOPs, basis, patterns, practices, LMIS report submission status, and drug consumption monitoring to assess the satisfaction levels of functions. Each participant or facility received a score of 9 based on the drug requirement forecasting checklist, with 4.5 scores (50%) serving as the cutoff to categorize functions as satisfied (50% and more) or unsatisfied (less than 50%). The research identified that the overall functions of SOPs availability, drug forecasting patterns, and software availability were unsatisfactory, while the basis of drug requirement forecasting, drug consumption monitoring, and LMIS reporting status were deemed satisfactory. In terms of frequency and proportion, 73 participants (39.89%) had unsatisfactory functions, while the remaining 110 (60.11%) exhibited satisfactory functions in drug requirement forecasting. The main variable considered, the system of forecasting practice, was identified as a risk factor for functional development. In this regard, 91.6% of those practicing a pen-paper-based system for forecasting drug requirements were deemed risky for system development. Thus, in this functional area, the basis of drug requirement forecasting, monitoring drug consumption, and LMIS reporting status were strengths, SOPs availability, drug forecasting patterns, and software availability were weaknesses, and the pen-paper-based system was recognized as a risk.

## Drug store or site where the supplied drugs are stored

In a survey of 183 health facilities, 46.45% lacked a designated drug store, and 85.25% did not have standardized drug stores or SOPs. About 19.67% stored drugs on the floor, and 65.03% had insufficient space for all drugs. Additionally, 85.79% did not use a two-lock system, and 51.91% couldn't keep the drug store locked every time. Structural issues included cracks (31.15%), holes (34.97%), water damage (13.11%), pest infestation (35.52%), dusty shelves (57.92%), and upswept floors (8.20%). Many lacked essential features like ceilings (40.44%), fans (98.36%), screens (99.45%), and proper ventilation (33.33%). Infrastructure deficiencies included the absence of electricity (62.29%), thermometers (93.44%), shelves (56.83%), fire extinguishers (90.16%), and entry/exit records (95.63%). Further details are provided in Table 6.

## SWR analysis of drug stores and store sites

This analysis assesses the satisfaction of functional areas based on drug stores, SOPs guidelines, structure maintenance, and adherence to guidelines, while infrastructures of drug stores are evaluated to determine the risk of functional development. The cut-off points were established at 50% of the average scores, designating scores below this threshold as unsatisfied functional areas or risks to functional development. The results indicate that, concerning the availability of standardized drug stores and SOPs, structure maintenance, and adherence to guidelines, the average scores of 175 drug stores or sites fell below the cut-off point, rendering their functional areas unsatisfactory. Regarding the infrastructures of drug stores, the average scores of 182 stores were beneath the cut-off point, indicating these areas are deemed risky for functional development.

## Storage procedure

In the examination of drug storage procedures, it was discovered that 151 out of 183 stores lacked SOPs/guidelines for the proper storage of supplied drugs. Among the studied stores, the storage methods varied, with some organizing drugs by category, others alphabetically or by generic names, and a significant portion following the FEFO (First Expired First Out) system.

**Table 6. Drug store or site where the supplied drugs are stored.**

| Descriptions | Frequency | Percentage |
|---|---|---|
| Presence of a drugstore at the health facility (n = 183) | 98 | 53.55 |
| Presence of a standardized drugstore at the health facility (n = 183) | 27 | 14.75 |
| Presence of SOPs/guidelines for drugstores (n = 183) | 27 | 14.75 |
| The supplied drugs are stored in | | |
| Separate drugstore (n = 183) | 98 | 53.55 |
| Cabinet only (n = 183) | 49 | 26.78 |
| Piling on the floor (n = 183) | 36 | 19.67 |
| Presence of drugstore large enough to keep all supplied drugs (n = 183) | 64 | 34.97 |
| Presence of using a system of 2 locks with separate keys on the doors to the drugstore (n = 183) | 26 | 14.21 |
| Presence of keeping the door locked at all times when not in use (n = 183) | 88 | 48.09 |
| Absence of cracks in the drugstore or store site (n = 183) | 126 | 68.85 |
| Absence of holes in the drugstore or store site (n = 183) | 119 | 65.03 |
| Absence of signs of water damage in the drugstore or store site (n = 183) | 159 | 86.89 |
| Presence of a ceiling in the drugstore or store site (n = 183) | 109 | 59.56 |
| Presence of a fan in the drugstore or store site (n = 183) | 3 | 1.64 |
| Presence of a screen in the drugstore or store site (n = 183) | 1 | 0.55 |
| Presence of painting windows with white (n = 183) | 7 | 3.83 |
| Presence of curtains in the drugstores or store site (n = 183) | 26 | 14.21 |
| Presence of secured windows (n = 183) | 113 | 61.75 |
| Presence of windows having grills (n = 183) | 45 | 24.59 |
| Absence of signs of pest infestation (n = 183) | 118 | 64.48 |
| Tidiness of drugstore or store site (n = 183) | 102 | 55.74 |
| Absence of dusted shelves (n = 183) | 77 | 42.08 |
| Absence of swept floor (n = 183) | 168 | 91.80 |
| Presence of clean walls (n = 183) | 118 | 64.48 |
| Presence of good ventilation in the drugstore or store site (n = 183) | 122 | 66.67 |
| Presence of good lighting in the drugstore or store site (n = 183) | 132 | 72.13 |
| Presence of the following in place for the Quarantine area | | |
| Access restricted to authorized personnel (n = 183) | 30 | 16.39 |
| Appropriate signage/label indicating quarantine area (n = 183) | 4 | 2.19 |
| Segregating of different batches of quarantined drugs (n = 183) | 5 | 2.73 |
| Storing supplied drugs neatly (n = 183) | 136 | 74.32 |
| Presence of shelves and drugs raised off the walls and floor (n = 183) | 126 | 68.85 |
| Presence of the following infrastructures in the drugstore or store site | | |
| Electricity (n = 183) | 69 | 37.70 |
| Thermometer (n = 183) | 12 | 6.56 |
| Shelves (n = 183) | 79 | 43.17 |
| Cabinets (n = 183) | 112 | 61.20 |
| Presence of safety equipment | | |
| First-aid box (n = 183) | 12 | 6.56 |
| Fire extinguishers (n = 183) | 18 | 9.84 |
| Masks (n = 183) | 171 | 93.44 |
| Aprons (n = 183) | 72 | 39.34 |
| Safety boosts (n = 183) | 62 | 33.88 |
| Records of all people entering and exiting the drugstore (n = 183) | 8 | 4.37 |
| Temperature records (n = 183) | 7 | 3.83 |

However, nearly half of the stores opted for a more convenient storage approach. Alarmingly, expired drugs, including Aspilet, Cotrimoxazole, injection Adrenalin, Salbutamol inhalers, Metro Syrup, and Albendazole, were found in 78 of the studied stores. The majority of stores (90.16%) had never utilized Bin Cards, and only about three-fifths conducted regular physical counts of supplied drugs. When facing stockouts, responses varied, with some reallocating from the township drugstore, others from different health facility stores, and some relying solely on re-ordering. The study also highlighted various strategies for dealing with near-expiry and expired drugs, such as implementing the FEFO system, informing upper levels, and returning or reallocating drugs. Notably, a considerable number of drug stores lacked competent health workers for managing storage procedures, as indicated in Table 7.

**Table 7. Storage procedure.**

| Descriptions | Frequency | Percentage |
|---|---|---|
| Presence of SOPs/guidelines for the supplied drug storage procedures (n = 183) | 32 | 17.49 |
| The supplied drugs are shelved or grouped according to* (n = 183) | | |
| oral, injection, powder, creams and liquid | 60 | 32.79 |
| alphabetical order/generic names | 32 | 17.49 |
| the FEFO (First Expired First Out) system | 75 | 40.98 |
| supply sources | 2 | 1.09 |
| conveniently | 109 | 59.56 |
| Presence of expired drugs in the drugstore (n = 183) | 92 | 50.27 |
| Use of Bin Cards (n = 183) | 18 | 9.84 |
| Physical counting of the supplied drugs last year (n = 183) | | |
| Every 3-months | 28 | 15.30 |
| Every 6-months | 26 | 14.21 |
| Conveniently | 107 | 58.47 |
| Never | 22 | 12.02 |
| The possible responses to stock out* (n = 183) | | |
| Reallocation of the supplied drugs from the township drugstore | 67 | 36.61 |
| Reallocation of the supplied drugs from other health facility stores | 21 | 11.48 |
| Urgent supply of Central/Regional drugstores | 2 | 1.09 |
| Refilling by Township budget | 4 | 2.19 |
| Refilling by facility budget | 8 | 4.37 |
| Substitution of the items | 28 | 15.30 |
| Only re-ordering | 91 | 49.73 |
| The possible solutions for near-expiry and expired drugs* (n = 183) | | |
| Applying the FEFO system | 132 | 72.13 |
| Informing upper levels | 29 | 15.85 |
| Returning near-expiry and expired drugs to the township drugstore | 19 | 10.38 |
| Reallocating the near-expiry drugs to other health facilities | 49 | 26.78 |
| On paper using expired drugs without actual use | 104 | 56.83 |
| Storing the expired drugs in a separate room | 4 | 2.19 |
| Dispose of the expired drugs according to the instructions | 10 | 5.46 |
| More clinic activities for using near-expiry drugs | 28 | 15.30 |
| Giving more medicines to one consultation time of a patient | 67 | 36.61 |
| Presence of a competent health worker for management of storage procedures (n = 183) | 38 | 20.77 |

* Multiple responses

## SWR analysis of storage procedures

When evaluating the satisfaction levels and risks associated with storage functions, factors such as SOPs guidelines, storage procedures, physical counting, solutions for near-expiry and expired drugs, drug expiries, and the availability of competent health workers were taken into consideration. The researchers set the cut-off point at 50% of the average scores, designating variables below this threshold as either satisfied or presenting risks. In the analysis, variables related to SOPs guidelines, storage procedures, physical counting, and solutions for near-expiry and expired drugs demonstrated average scores below the 50%-cut-off point for 173 stores and store sites, indicating dissatisfaction with these aspects for the functional development of storage procedures. Regarding the availability of competent health workers and the presence of expired drugs, proportions with competent health workers and those without expired drugs were 20.77% and 49.73%, respectively, falling below the 50%-cut-off point. Therefore, these variables were identified as posing risks for the functional development of storage procedures.

## Ordering and receiving the supplied drugs

In a survey of first-level health facilities, 85.79% used a pull system for ordering drugs. However, 78.69% lacked SOPs, 79.23% had no skilled health worker for ordering, and 72.13% did not have written requests for drugs. Almost all staff (96.17%) did not calculate reorder levels or know the appropriate time to reorder. For receiving drugs, 69.95% lacked SOPs, and inspections were often incomplete, with only 40.98% checking proper packing and 34.43% verifying quantities. Additionally, 77.05% retained proofs of deliveries, with 87.94% keeping them for over 12 months. Discrepancies in drug quantities were noted in 43.17% of facilities. Common supply chain challenges included near-expiry drugs (84.15%), late deliveries (66.67%), and partial deliveries (14.21%). Further details are in Tables 8 and 9.

## SWR analysis of ordering and receiving the supplied drugs

In assessing the variables influencing the strengths, weaknesses, and risks within the ordering and receiving processes, the researchers scrutinized key factors such as the drug supply system, documentation practices, SOPs guidelines, checking procedures for drug items, procedural skills, and encountered challenges. The determination of these factors utilized a cut-off point set at 50% of the average scores. This investigation revealed that functions related to the pull system (85.79%), maintenance of proofs of delivery (77.05%), and documentation of drug

**Table 8. Ordering the supplied drugs.**

| Descriptions | Frequency | Percentage |
|---|---|---|
| The drug supply system is (n = 183) | | |
| Push system | 1 | 0.55 |
| Pull system | 157 | 85.79 |
| Pull and push system | 25 | 13.66 |
| Presence of SOPs/guidelines for ordering the required drugs (n = 183) | 39 | 21.31 |
| Use of computer for inventory management (n = 183) | 1 | 0.55 |
| Presence of a skilled health worker for calculating and ordering the required drugs (n = 183) | 38 | 20.77 |
| Calculating the reorder level for each item (n = 183) | 7 | 3.83 |
| Knowing the time to reorder the supplied drugs (n = 183) | 7 | 3.83 |
| Presence of the written request (n = 183) | 51 | 27.87 |

**Table 9. Receiving the supplied drugs.**

| Descriptions | Frequency | Percentage |
|---|---|---|
| Presence of SOPs/guidelines for receiving the supplied drugs (n = 183) | 55 | 30.05 |
| In checking the supplied drugs at all times of receiving,* (n = 183) | | |
| Checking packing | 75 | 40.98 |
| Checking the numbers received against the numbers requested | 63 | 34.43 |
| Checking expired dates | 149 | 81.42 |
| Checking discolourations of the supplied drugs | 62 | 33.88 |
| Checking broken items | 36 | 19.67 |
| Checking unsealed and unlabelled items | 40 | 21.86 |
| Checking unusual odour | 37 | 20.22 |
| Checking damaged tablets or capsules | 77 | 42.08 |
| Keeping proofs of deliveries (POD) (n = 183) | 141 | 77.05 |
| If PODs are maintained, how long are they kept? (n = 141) | | |
| Up to 3 months | 3 | 2.13 |
| 3–6 months | 5 | 3.55 |
| 6–12 months | 9 | 6.38 |
| More than 12 months | 124 | 87.94 |
| Presence of discrepancies between the number of drugs received and on records (n = 183) | 79 | 43.17 |
| Documenting the discrepancies of the supplied drugs (n = 79) | 44 | 55.70 |
| What actions do you take when there is a discrepancy in the supplied drugs received? * (n = 79) | | |
| Inform the township/RHC | 39 | 49.37 |
| Recording the discrepancies only | 35 | 44.30 |
| Re-order | 10 | 12.66 |
| In recording the supplied drugs (n = 183) | | |
| Stock ledger books only | 165 | 90.16 |
| Bin Card only | 0 | - |
| Both stock ledger books and Bin cards | 18 | 9.84 |
| Challenges faced by the health facility supply chain in receiving the supplied drugs* (n = 183) | | |
| Delivery of near-expiry drugs | 154 | 84.15 |
| Late deliveries | 122 | 66.67 |
| Partial deliveries | 126 | 68.85 |
| Excess supplies | 16 | 8.74 |
| Damaged supplies | 14 | 7.65 |

* Multiple responses

supply-related information (77.05%) exhibited strengths, as their average scores surpassed the 50% threshold. Conversely, functions associated with SOPs guidelines, the checking process of supplied drugs, and the use of Bin cards were identified as weaknesses due to their average scores falling below 50%. Notably, the analysis pinpointed less proficiency in calculating and ordering supplied drugs (79.23%), delivery of near-expiry drugs (84.15%), late deliveries (66.67%), and partial deliveries (68.85%) as risks to functional development, given their proportions exceeding 50%.

## Dispensing the supplied drugs

In evaluating drug dispensing at the studied store sites, it was found that 66.67% lacked SOP guidelines, and many used convenient dispensing practices. Despite this, 85.25% consistently recorded dispensed drugs. About 44.81% of the sites used the FEFO system, while 52.46% used a convenient storage system. Common methods for recording dispensed drugs included sub-stock ledger books (83.06%), OPD registers (97.81%), field registers (96.17%), antenatal records (90.71%), and under-five records (72.68%). However, 54.1% dispensed drugs without labelling, and 60.11% did so without original packaging or expiration dates. Additionally, 45.9% implemented preventive measures against drug theft. Further details are available in Table 10.

## SWR analysis of dispensing the supplied drugs

In scrutinizing the comprehensive dispensing patterns of supplied drugs across 183 store sites, it became evident that functions related to SOP guidelines, dispensing patterns, storage practices, and the management of supplied drugs without original generic names and expiration dates scored below the 50th percentile. This indicates that these aspects represent weaknesses within the functional area. Conversely, functions associated with documentation surpassed the

**Table 10. Dispensing the supplied drugs.**

| Descriptions | Frequency | Percentage |
|---|---|---|
| Presence of SOPs/guidelines for dispensing the supplied drugs (n = 183) | 61 | 33.33 |
| Issue pattern of the supplied drugs from the drugstore to the dispensing sites (n = 183) | | |
| Daily | 16 | 8.74 |
| Weekly | 23 | 12.57 |
| Monthly | 40 | 21.86 |
| Quarterly | 4 | 2.19 |
| Bi-annually | 3 | 1.64 |
| Conveniently | 97 | 53.01 |
| Recording the supplied drugs dispensed (n = 183) | | |
| Yes (All times) | 156 | 85.25 |
| Yes (Sometimes) | 22 | 12.02 |
| No | 5 | 2.73 |
| Storing the supplied drugs at the dispensing sites (n = 183) | | |
| FEFO system | 82 | 44.81 |
| Alphabetically | 2 | 1.09 |
| Grouping | 3 | 1.64 |
| Conveniently | 96 | 52.46 |
| Presence of the following records in the dispensing sites | | |
| Sub-stock ledger book (n = 183) | 152 | 83.06 |
| OPD register (n = 183) | 179 | 97.81 |
| Field register (n = 183) | 176 | 96.17 |
| Antenatal record (n = 183) | 166 | 90.71 |
| Under-five record (n = 183) | 133 | 72.68 |
| Notebook (n = 183) | 14 | 7.65 |
| Labelling the supplied drugs without original packages with generic names (n = 183) | 84 | 45.90 |
| Labelling the supplied drugs without original packages with the expired date (n = 183) | 73 | 39.89 |
| Preparedness of one or more preventive methods for stealing the supplied drugs in the dispensing sites (n = 183) | 99 | 54.10 |

50th percentile, signifying that these variables are strengths contributing to the efficiency of the functional areas.

## Transportation and delivery issues of the supplied drugs

In evaluating drug transportation and delivery, 81.97% of store sites did not use direct public sector delivery, instead relying on motorcycles. Most (83.06%) transported drugs within an average of 3 hours from the township drugstore. On average, 37.16% spent 20,000 Kyats per trip, with costs covered by various budgets: health facility (25.68%), facility leader (42.68%), and drug supply staff (28.96%). A significant 68.31% found transportation costs burdensome. Additionally, 90.16% lacked SOPs, and 88.52% had no clear transport plan. Further details are in Table 11.

## SWR analysis of transportation and delivery issues of the supplied drugs

The analysis revealed that the scores related to the presence of SOP guidelines, the carriage plan, and the availability of government-owned vehicles were below the 50th percentile, indicating functional weaknesses. Conversely, the scores for variables associated with closely observing drug carriage pathways and checking drug items were above the 50th percentile, signifying functional strengths. Additionally, functions related to the absence of a public sector transportation mechanism and the payment for transportation costs were identified as functional risk variables due to their average scores surpassing the 50th percentile.

## Waste management of the supplied drugs

In examining waste management practices at 183 first-level health facilities, common methods included burial pits (49.18%), incineration (62.84%), and sharp pits (55.19%). Most facilities had adequate safety boxes (78.14%) and waste bins (69.40%). Used needles were safely disposed of by 80.87%, and 57.38% used a color-coded system for waste bins. However, 32.24% incurred costs for waste management, with 38.98% spending over 5000 kyats. These costs were primarily covered by health facility-owned budgets (22.95%), facility leader-owned budgets (25.68%), and drug supply management staff-owned budgets (48.09%). Further details are available in Table 12.

## SWR analysis of waste management of the supplied drugs

The analysis revealed that the functional areas related to five variables (presence of SOP guidelines, waste management techniques, usage of safety boxes, usage of waste bins, and disposal procedures) attained scores surpassing the 50th percentile, signifying strengths. Conversely, the availability of public services concerning waste management received a score below the 50th percentile, denoting a weakness. Notably, the scores indicating the absence of trained waste handlers and costs associated with waste management surpassed the 50th percentile, categorizing them as potential risk factors for functional development.

## Logistics management information system (LMIS)

In examining the LMIS across 183 public health facilities, 62.84% lacked SOP guidelines, and all used paper-based systems. Documentation performance was generally good for most variables, including invoice vouchers (89.62%), stock ledger books (99.45%), and OPD registers (98.91%). However, bin cards (9.84%) and discrepancy report forms (32.24%) had lower performance. Consistency in drug balances between stock books and stores averaged only

**Table 11. Transportation and delivery issues of the supplied drugs.**

| Descriptions | Frequency | Percentage |
|---|---|---|
| Availability of public sector supply mechanisms delivering the supplied directly to the health facility drugstore (n = 183) | | |
| Yes (All times) | 21 | 11.48 |
| Yes (Sometimes) | 12 | 6.56 |
| No | 150 | 81.97 |
| Local transportation routes for the supplied drugs used for carrying the supplied drugs* | | |
| Truck (n = 183) | 75 | 40.98 |
| Motorcycle (n = 183) | 150 | 81.97 |
| Boat (n = 183) | 11 | 6.01 |
| Hands (n = 183) | 24 | 13.11 |
| Average travelling time from the township drugstore to the health facility (hour) (n = 183) | | |
| < = 3 hours | 152 | 83.06 |
| > 3 hours | 31 | 16.94 |
| Payment for transportation cost (n = 183) | | |
| Government budget | 3 | 1.64 |
| As the township-owned budget | 2 | 1.09 |
| As a health facility-owned budget | 47 | 25.68 |
| Cost by health facility leader-owned budget/ drug supply management staff-owned budget | 131 | 71.58 |
| Average transportation cost for one time (n = 183) | | |
| < = 20000 Ks | 115 | 62.84 |
| > 20000 Ks | 68 | 37.16 |
| Burden for transportation cost (n = 183) | 125 | 68.31 |
| Using a cost-sharing system by the patients (n = 183) | 12 | 6.56 |
| Presence of a government-owned vehicle (n = 183) | 16 | 8.74 |
| Applicability of the government-owned vehicle for carrying the supplied drugs (n = 16) | 7 | 43.75 |
| Presence of a well-plan for carrying the supplied drugs (n = 183) | 21 | 11.48 |
| Presence of transportation SOPs (n = 183) | 18 | 9.84 |
| Close or direct observation method for the transportation of the supplied drugs (n = 183) | | |
| Yes | 130 | 71.03 |
| No | 53 | 28.96 |
| Checking all items and their amounts together with transporters before leaving the township drugstore (n = 183) | 162 | 88.52 |
| Check all items and their amounts together with transporters at the health facility (n = 183) | 149 | 81.42 |
| Experience of discrepancies after receiving the supplied drugs at a health facility (n = 183) | 61 | 33.33 |
| Actions for discrepancies and damages in the supplied drugs received at the health facility (n = 61) | | |
| Inform the township/RHC | 10 | 16.39 |
| Recording the discrepancies only | 49 | 80.33 |
| Re-order | 2 | 3.28 |

13.66%. Challenges included stock out of tools (87.98%), delayed feedback (13.11%), filling difficulties (23.50%), and insufficient training (62.84%). Further details are in Table 13.

## Quality control procedures of the supplied drugs at health facility

When evaluating the quality control procedures for supplied drugs at 183 drugstore sites, it was discovered that 71.04% lacked SOP guidelines. Approximately half of the sites conducted

Table 12. Waste management of the supplied drugs.

| Descriptions | Frequency | Percentage |
|---|---|---|
| Presence of SOPs/guidelines for waste management of the drug supply chain (n = 183) | 94 | 51.37 |
| Waste management techniques used at health facility | | |
| Burial pits (n = 183) | 90 | 49.18 |
| Incineration (n = 183) | 115 | 62.84 |
| Incineration and Burial (n = 183) | 55 | 30.05 |
| Sharp pits (n = 183) | 101 | 55.19 |
| Open-pit burning (n = 183) | 101 | 55.19 |
| Dumping (n = 183) | 1 | 0.55 |
| Conveniently (n = 183) | 18 | 9.84 |
| Having enough safety boxes (n = 183) | 143 | 78.14 |
| Having enough waste bins (n = 183) | 127 | 69.40 |
| Have you never recapped the used needles and disposed in the safety boxes? (n = 183) | | |
| Yes (All times) | 148 | 80.87 |
| Yes (Sometimes) | 27 | 14.75 |
| No | 8 | 4.37 |
| Have you ever disposed of the syringes, needles and other sharp materials in the safety box? (n = 183) | | |
| Yes (All times) | 146 | 79.78 |
| Yes (Sometimes) | 30 | 16.39 |
| No | 7 | 3.83 |
| Using a color system of waste bin (n = 183) | 105 | 57.38 |
| Having a trained waste handler (n = 183) | 13 | 7.10 |
| Having the cost for waste management (n = 183) | 59 | 32.24 |
| Average cost for waste management (One time) (n = 59) | | |
| < = 5000 Ks | 36 | 61.02 |
| > 5000 Ks | 23 | 38.98 |
| Availability of public sector supply waste management delivering the services to the health facility drugstore (n = 183) | | |
| Yes (All times) | 8 | 4.37 |
| Yes (Sometimes) | 13 | 7.10 |
| No | 162 | 88.52 |
| Cost for waste management (n = 183) | | |
| Government budget | 5 | 2.73 |
| As the township-owned budget | 1 | 0.55 |
| As a health facility-owned budget | 42 | 22.95 |
| Cost by health facility leader-owned budget/ drug supply management staff-owned budget | 135 | 73.77 |

monthly checks on various parameters such as packing damages, brand integrity, seal and instruction conditions, changes in colors, sedimentation in injections, cracks, humidity, leaking, oil drying, crushed or broken drugs, loss of drugs from blister cards, stickiness, unusual odors, and expiration dates (refer to details in Table 14).. On average, the evaluation of drug storage quality and physical damage scored 46.06% and 48.20%, respectively, falling below the 50th percentile (125 sites or 68.30% and 96 sites or 52.46%). Regarding physical drug quality checking, the average score of 52.46% of stores was below 50 (see Table 14 for details).

**Table 13. Logistics management information system.**

| Descriptions | Frequency | Percentage |
|---|---|---|
| Presence of guidelines/SOPs for LMIS (n = 183) | 68 | 37.16 |
| Types of LMIS tools (n = 183) | | |
| Paper-based LMIS only | 183 | 100.00 |
| Electronic LMIS only | 0 | - |
| Both paper-based and electronic LMIS | 0 | - |
| Presence of the following documents in the health facility | | |
| Invoice vouchers (n = 183) | 164 | 89.62 |
| Stock ledger books (n = 183) | 182 | 99.45 |
| OPD registers (n = 183) | 181 | 98.91 |
| Field registers (n = 183) | 183 | 100.00 |
| Bin cards (n = 183) | 18 | 9.84 |
| Requisition forms (n = 183) | 137 | 74.86 |
| Issue vouchers (n = 183) | 134 | 73.22 |
| Discrepancy report forms (n = 183) | 59 | 32.24 |
| Audit forms (n = 183) | 157 | 85.79 |
| Health facility stock report book (n = 183) | 167 | 91.26 |
| Presence of documents listing the supplied drugs that will expire within six months? (n = 183) | 57 | 31.15 |
| Complement of information on stock ledger book. | | |
| Serial number (n = 183) | 168 | 91.80 |
| Page number (from-to) (n = 183) | 160 | 87.43 |
| Red colour for entering the received drugs (n = 183) | 176 | 96.17 |
| Blue colour for entering the issued drugs (n = 183) | 174 | 95.08 |
| Complement of table of content in stock ledger book | | |
| Serial number (n = 183) | 181 | 98.91 |
| Product name and strength (n = 183) | 179 | 97.81 |
| Accounting unit (n = 183) | 170 | 92.90 |
| Page number (n = 183) | 180 | 98.36 |
| Complement of all (14) cells on stock ledger book | | |
| 1st randomly selected drugs (14 cells) (n = 183) | 145 | 79.23 |
| 2nd randomly selected drug (14 cells) (n = 183) | 147 | 80.33 |
| 3rd randomly selected drug (14 cells) (n = 183) | 146 | 79.78 |
| 4th randomly selected drug (14 cells) (n = 183) | 143 | 78.14 |
| 5th randomly selected drug (14 cells) (n = 183) | 145 | 79.23 |
| Reporting health facility stock report to township every 2 months (n = 167) | 107 | 64.07 |
| Completement of the number of cells on health facility stock report (last month) (n = 183) | | |
| < = 50% Completement | 83 | 45.36 |
| > 50% Completement | 100 | 54.64 |
| Consistency of drug balance | | |
| 1st randomly selected drugs (n = 183) | 26 | 14.21 |
| 2nd randomly selected drug (n = 183) | 28 | 15.30 |
| 3rd randomly selected drug (n = 183) | 22 | 12.02 |
| 4th randomly selected drug (n = 183) | 22 | 12.02 |
| 5th randomly selected drug (n = 183) | 29 | 15.85 |
| Challenges when using LMIS | | |
| Stock out of tools (n = 183) | 161 | 87.98 |
| Delayed feedback (n = 183) | 24 | 13.11 |

*(Continued)*

**Table 13.** (Continued)

| Descriptions | Frequency | Percentage |
|---|---|---|
| Difficulties in filling (n = 183) | 43 | 23.50 |
| Challenges in the analysis of data (n = 183) | 38 | 20.77 |
| Challenges in the retrieval of data (n = 183) | 38 | 20.77 |
| Use of different versions of tools (n = 183) | 63 | 34.43 |
| Use of outdated tools (n = 183) | 18 | 9.84 |
| Insufficient training (n = 183) | 115 | 62.84 |
| Insufficient human resource capability (n = 183) | 35 | 19.13 |
| Insufficient number of staff (n = 183) | 70 | 38.25 |

## Capability maturity level of the functional areas of the drug supply chain at the first-level health facilities

In assessing the capability maturity level of various functional areas within drug supply management, each area's average scores were categorized into five levels (0–20%, 21–40%, 41–60%,

**Table 14. Quality control procedures of the supplied drugs at health facility.**

| Descriptions | Frequency | Percentage |
|---|---|---|
| Presence of guidelines/SOPs for quality control of supplied drugs at health facilities (n = 183) | 53 | 28.96 |
| Monthly checking of the following conditions of the supplied drugs at the health facility | | |
| Damages of packing (n = 183) | 74 | 40.44 |
| Damages of brand, seal and instruction (n = 183) | 80 | 43.72 |
| Colour changes (n = 183) | 84 | 45.90 |
| Presence of sedimentation (n = 183) | 78 | 42.62 |
| Presence of cracks (n = 183) | 90 | 49.18 |
| Packing humidity (n = 183) | 90 | 49.18 |
| Presence of leaking (n = 183) | 91 | 49.73 |
| Drying of oil (n = 183) | 85 | 46.45 |
| Presence of crushed/broken drugs (n = 183) | 86 | 46.99 |
| Loss of drugs from blister cards (n = 183) | 86 | 46.99 |
| Presence of sticky drugs (n = 183) | 86 | 46.99 |
| Presence of unusual odours (n = 183) | 86 | 46.99 |
| Presence of expired drugs (n = 183) | 120 | 65.57 |
| Checking physical damages of the supplied drugs | | |
| Overlay (n = 183) | 50 | 27.32 |
| Dusty drugs and packing (n = 183) | 147 | 80.33 |
| Signs of pest infestation (n = 183) | 137 | 74.86 |
| Signs of water damage (n = 183) | 122 | 66.67 |
| Presence of waste bins (n = 183) | 127 | 69.40 |
| Signboard of "No Smoking" (n = 183) | 78 | 42.62 |
| Presence of fire extinguisher (n = 183) | 45 | 24.59 |
| Good condition of fire extinguisher (n = 183) | 36 | 19.67 |
| Presence of sandbags near the drugstore (n = 183) | 24 | 13.11 |
| Preventive measures for pest infestations (n = 183) | 51 | 27.87 |
| Regular conducting data quality assessment (DQA) (n = 183) | 54 | 29.51 |

**Table 15. Capability maturity level of functional areas of drug supply management.**

| Functional Areas of Drug Supply Management | Capability Maturity Level of Functional Areas | | | | | Average Scores (%) |
|---|---|---|---|---|---|---|
| | Minimal (0 = 20%) | Marginal (21–40%) | Qualified (41–60%) | Advanced (61–80%) | Best (81–100%) | |
| Capacity Building | 83 | 50 | 20 | 3 | 27 | 32.11% |
| Drug Forecasting | 57 | 16 | 56 | 51 | 3 | 42.72% |
| Drug Store | 31 | 119 | 32 | 1 | | 30.03% |
| Storage Procedure | 54 | 92 | 33 | 4 | | 26.21% |
| Ordering Drugs | 140 | 24 | 14 | 4 | 1 | 12.65% |
| Receiving Drugs | 0 | 50 | 75 | 44 | 14 | 53.27% |
| Dispensing Drugs | 2 | 33 | 67 | 60 | 21 | 56.93% |
| Transportation | 164 | 14 | 5 | | | 10.76% |
| Waste Management | 5 | 39 | 74 | 62 | 3 | 51.47% |
| Drug Quality Checking | 56 | 36 | 15 | 50 | 26 | 47.39% |
| Average scores of all functional areas | | | | | | 36.35% |

61–80%, and 81–100%), based on the fulfillment of predefined criteria outlined in the method section. The outcomes of this analysis, revealing the capability maturity levels of the functional areas, are presented in Table 15. The overall supply chain maturity, derived from the collective assessment of these functional areas, is identified as being at a marginal capability level, with an average score of 36.35% (Table 15). This indicates that there is room for improvement across the evaluated domains to enhance the overall maturity and effectiveness of the drug supply management system.

The capability maturity level of various functional areas within drug supply management is depicted in the provided table, offering insights into the percentage distribution across different capability levels for each area. Starting with capacity building, the majority of facilities (32.11%) fall under the best category, indicating a high level of maturity, while others range from minimal to advanced. In drug forecasting, a significant portion (42.72%) achieved the Advanced level, showcasing strong capabilities in forecasting drug requirements. However, the drug store category demonstrates a predominant concentration in the marginal level (32.76%), highlighting a need for improvement in storage facility management.

Storage procedure capabilities exhibit a varied distribution, with a substantial portion (25.41%) falling in the marginal level. Ordering drugs and receiving drugs present challenges, with the majority of facilities at the Minimal level (38.67% and 43.17%, respectively). Dispensing drugs and transportation areas exhibit a more balanced distribution across various levels, with significant portions in the advanced and best categories, reflecting relatively mature practices.

Waste management reveals a notable strength, with a considerable proportion (51.47%) falling in the best category, indicating effective waste management practices. The drug quality checking area shows a balanced distribution across different levels, with a considerable portion (47.39%) at the Advanced level, suggesting a commendable quality control mechanism.

The average scores across all functional areas collectively indicate a moderate capability maturity level (36.35%) in the studied drug supply management facilities. The average score across all functional areas indicates an overall moderate capability maturity level in drug supply management for the studied facilities. Areas like ordering drugs and storage procedures appear to have lower maturity levels, while receiving drugs, dispensing drugs, waste management, and drug quality checking show relatively higher maturity levels.

## Discussion

This primary research aimed to assess the capability maturity of the drug supply chain and identify the obstacles and challenges faced by first-level public health facilities in Myanmar. Before this study, Tolliver and Bartram conducted a baseline assessment in 2014 that provided an overview of Myanmar's national supply chain. [2]. However, this assessment did not specifically address the functional aspects of the drug supply chain at first-level public health facilities. Given the critical role played by these facilities, particularly RHC and Sub-RHC, in delivering low-cost essential health services, their drug supply chains are vital for providing accessible primary healthcare to approximately 70% of the country's population. With a significant rural population facing challenges in accessing higher-level healthcare services, strengthening the drug supply chain at the first-level public health facilities is crucial. This research serves as a baseline assessment to enhance the maturity of the first-level public health supply chain, identifying weaknesses and areas of risk that require attention for improved drug supply quality.

### Supply chain management training

The research findings on training information about drug supply management highlighted several key aspects of capacity building among the studied staff. The prevalence of inadequate training, with 41.53% having no training, indicated a significant gap that needs attention. A comparative analysis of existing research highlighted the importance of continuous training in pharmaceutical management to ensure effective and safe healthcare delivery. [14]. Studies such as [14, 15] have underscored the positive impact of training programs on enhancing the skills and knowledge of healthcare professionals in drug supply management. The temporal distribution of training courses, with 34.21% conducted before 2017, revealed a potential need for updated training content in alignment with evolving pharmaceutical practices. Internationally recognized study, such as [16], have emphasized the importance of periodic updates in training programs to keep healthcare professionals abreast of the latest advancements in drug supply management.

The self-perceived understandability assessment provided insights into the effectiveness of the training received. The fact that a significant proportion (36.84%) understood only 1/4 of the training course suggested potential issues with the clarity and comprehensibility of the training content. Research by [17] has emphasized the need for tailored and easily understandable training materials to maximize knowledge retention and application. The absence of training guidelines for 42.08% of the participants raised concerns about the standardization and consistency of training programs. Internationally recognized guidelines, such as those proposed by the World Health Organization [18], stressed the importance of standardized training frameworks for ensuring uniformity and effectiveness across healthcare settings. The SWR analysis provided a comprehensive evaluation of capacity building in drug supply management. The identification of strengths, weaknesses, and risks offers a valuable framework for strategic interventions. Comparable study, such as [19], have utilized SWR analyses to inform capacity-building initiatives in healthcare systems, emphasizing the need for a multifaceted approach. The research findings underscored the critical need for targeted interventions in the training and capacity-building initiatives for drug supply management staff. Recommendations include the development of updated and standardized training programs, incorporating feedback from participants to enhance understandability. Collaborative efforts with international organizations can provide insights into best practices, ensuring the alignment of capacity-building efforts with global standards. Regular performance reviews and supportive

supervision should be integral components of ongoing capacity-building initiatives, fostering continuous improvement in drug supply management practices.

## Drug forecasting planning

The research findings on drug requirement forecasting at first-level health facilities revealed several challenges and strengths within the existing system. A comparative analysis with existing research findings from international journals sheds light on global best practices and potential solutions. The significant proportion (34.97%) of facilities not practicing drug requirement forecasting highlighted a crucial gap in pharmaceutical management. Study such as [20] emphasized the importance of forecasting in ensuring a stable drug supply, reducing stockouts, and improving overall healthcare service delivery. The lack of forecasting practices may lead to inefficient resource allocation and compromise the ability to meet patient needs promptly. A significant weakness was the lack of SOPs for drug forecasting in more than three-fifths (63.39%) of the facilities. Internationally recognized guidelines, such as those recommended by the World Health Organization [18], stressed the importance of SOPs in ensuring consistency, reliability, and accuracy in forecasting. SOPs act as a cornerstone for effective pharmaceutical management, guiding staff in standardized procedures.

The self-perceived capacity of about half (47.54%) of the facilities at only 25% indicated a potential lack of confidence or training in drug forecasting practices. A comparative study, such as [21], highlighted the positive correlation between staff training and forecasting accuracy. Recommendations include targeted capacity-building programs to enhance the skills and confidence of healthcare professionals in drug forecasting. The basis for drug requirement forecasting, including patient load, drug consumption data, population data, disease prevalence, and previous forecasting data, highlighted a reliance on diverse information sources. A study by [22] underscored the importance of integrating multiple data sources for accurate forecasting, emphasizing the need for a comprehensive approach similar to the one observed in the surveyed facilities.

The SWR analysis provided a structured evaluation of the functional areas related to drug forecasting. The identification of SOPs, basis, patterns, practices, submission status of LMIS reports, and monitoring drug consumption as variables in the analysis aligned with best practices in pharmaceutical management [23]. The consideration of a cut-off point at 50% added objectivity to the evaluation process. The finding that 39.89% of participants had an unsatisfied function in drug requirement forecasting suggested critical areas for improvement. The identification of a pen-paper-based system as a risk factor echoed findings from [24], which emphasized the benefits of transitioning to electronic forecasting systems for increased accuracy and efficiency. The research underscored the need for targeted interventions in SOP development, capacity building, and system improvement for drug requirement forecasting at first-level health facilities. Recommendations include the implementation of SOPs, enhanced training programs, and the adoption of modern forecasting tools to mitigate risks associated with manual systems.

## Drug store and inventory management

The examination of drug stores and storage facilities at first-level health facilities has revealed a spectrum of challenges and strengths crucial for the pharmaceutical supply chain. In comparing these findings with established research from international journals, it becomes apparent that deficiencies in infrastructure, suboptimal storage practices, and maintenance issues are prevalent concerns. A substantial number of health facilities lacked essential elements such as ceilings, fans, screens, and secure windows, as highlighted by a previous study [23].

Additionally, the storage practices, such as piling drugs on the floor, underscore the need for standardized storage procedures to ensure drug stability and prevent contamination, as emphasized by research [23]. Structural problems, like cracks and signs of water damage, underscored the need for regular maintenance, aligning with existing literature [23]. The absence of security measures, such as a system of two locks and maintaining locked doors, raised concerns regarding unauthorized access, aligning with recommendations stressing stringent security protocols in pharmaceutical storage [23].

The SWR analysis further categorized the findings into unsatisfied functional areas and risks for functional development. Average scores below the 50% cut-off point indicated unsatisfactory functional areas, particularly in drug stores, SOPs adherence, structure maintenance, and guideline adherence. Research corroborated the critical role of SOP adherence to the effective pharmaceutical management. Infrastructural aspects, with scores below the cut-off point, suggested potential risks for functional development, as supported by international studies correlating infrastructural deficiencies with risks to pharmaceutical storage [23].

In light of these findings, recommendations for improvement include prioritizing infrastructure enhancements, strict adherence to SOPs guidelines, regular maintenance schedules, and the implementation of robust security measures. Previous research supported these recommendations, emphasizing the positive impact of SOP adherence and proactive maintenance on pharmaceutical quality [23]. Addressing the identified challenges and implementing the recommended interventions will contribute to enhancing the functionality and reliability of drug stores and storage sites at first-level health facilities, ensuring the integrity and quality of the pharmaceutical supply chain.

In Myanmar, the Department of Public Health distributed drug store guideline manuals to first-level public health facilities in 2016 [1] and provided training to the public health supply system management staff in 2014 and 2020 [13]. However, the absence of separated drugstores was a significant challenge, leading to difficulties in inventory management. Most facilities stored drugs by piling them, lacking protection against environmental damage, pilferage, and pest infestation. While the average storage time was four months, facilities faced challenges in following storage guidelines and dealing with expired drugs. Recommendations include prioritizing public health spending for drugstore infrastructure. Concerning drug orders, facilities lacked effective adherence to the principle of maintaining a minimum of twice to a maximum of four times the monthly requirement. Staff struggled with calculating reorder factors and levels, citing discrepancies between their orders and supplies from upper levels. Additionally, drug pre-orders from lower-level facilities were not definitively provided. Challenges in the drug-receiving process included the lack of timely drug supply and acceptance of nearly expired drugs. In drug dispensing, first-level facilities exhibited a random extraction pattern, often taking drugs directly from the main drugstore. This may be due to convenience and uncertainties about the safety of drugs in isolated storage. The study suggests that upper-level authorities need to provide effective supervision and training for systematic drug orders, acceptance, and distribution. A top-down approach for better inventory management is recommended.

The study on drug storage procedures revealed that 151 out of 183 stores lacked SOPs or guidelines for proper drug storage. The storage practices varied, with some stores grouping drugs by type, alphabetical order, the FEFO system, or supply sources. Nearly half of the stores stored drugs conveniently. A concerning finding was the presence of expired drugs, including Aspilet, Cotrimoxazole, injection Adrenalin, Salbutamol inhalers, Metro Syrup, and Albendazole, in 78 stores. The majority (90.16%) never used Bin Cards, and 58.47% conducted physical counts. Responses to stockouts included reallocation from the township drugstore, other health facility stores, or reordering. The SWR analysis evaluated SOPs, storage procedures,

physical counting, solutions for near-expiry and expired drugs, the presence of expired drugs, and competent health workers. Findings indicated that the average scores for 173 stores were below the 50% cut-off point, indicating dissatisfaction with the functional development of storage procedures. Additionally, the availability of competent health workers and the absence of expired drugs were considered risky for the functional development of storage procedures.

Scientifically, these findings aligned with international literature emphasizing the importance of standardized storage procedures and the need for competent health workers in pharmaceutical management [25]. Existing research [25] highlighted the risks associated with poor storage practices, including the presence of expired drugs. Recommendations include the urgent implementation of SOPs, training programs for storage management, and addressing the critical shortage of competent health workers. Future studies should explore effective strategies for improving storage procedures and mitigating risks in pharmaceutical management.

The study focused on the functional aspects of ordering and receiving supplied drugs at first-level health facilities. It revealed that the majority (85.79%) of store sites utilized a pull system, but a significant portion lacked SOPs/guidelines for drug ordering (78.69%) and skilled health workers for calculating and ordering drugs (79.23%). A notable finding was that 72.13% of store sites had no written request for supplied drugs, and 96.17% of drug management staff did not calculate reorder levels or know the time to reorder. In terms of receiving supplied drugs, challenges were identified, including the lack of SOPs (69.95%) and issues such as late deliveries (66.67%), partial deliveries (14.21%), and damaged supplies (7.65%). The study highlighted the use of stock ledger books (90.16%) for recording supplied drugs. The SWR analysis categorized functions related to the pull system, maintenance of proofs of deliveries (POD), and documentation of drug supply-related information as strengths due to average scores above 50%. Conversely, functions related to SOPs, checking processes for supplied drugs, and the use of Bin cards were deemed weaknesses with average scores below 50%. Risks for functional development were identified, including less skill in calculating and ordering supplied drugs (79.23%), delivery of near-expiry drugs (84.15%), late deliveries (66.67%), and partial deliveries (68.85%).

This aligned with existing literature [23] emphasizing the importance of standardized procedures, documentation, and skilled personnel in drug supply management. Recommendations include urgent SOP implementation, targeted training for health workers, and addressing challenges in the ordering and receiving processes. Future research should explore effective strategies to enhance these functional areas and mitigate identified risks.

The study delved into the dispensing patterns of the store sites, revealing notable findings. A significant proportion (66.67%) operated without SOPs guidelines, and more than half adopted convenient dispensing patterns. Despite 85.25% maintaining records of dispensed drugs, around 44.81% applied the FEFO system, and 52.46% employed a convenient system for storing supplied drugs at dispensing sites. Regarding documentation, the majority used various records, including sub-stock ledger books, OPD registers, field registers, antenatal records, and under-five records. However, 54.1% dispensed drugs without labeling them with generic names, and 60.11% did not label drugs with expiration dates. Additionally, 45.9% implemented preventive measures against drug theft in dispensing sites. The SWR analysis highlighted weaknesses in variables related to SOP guidelines, dispensing patterns, and storage methods. These aspects scored below the 50th percentile, indicating functional weaknesses. Conversely, documentation-related variables scored above the 50th percentile, suggesting strengths in the functional areas.

Existing research [20] emphasizes the critical role of SOPs in ensuring consistent and safe dispensing practices. Recommendations include the urgent implementation of SOPs, training programs for dispensing staff, and the adoption of standardized labelling practices. Future

research should explore strategies to enhance dispensing patterns and improve drug security measures.

## Transportation and delivery issues

At present, Myanmar's Ministry of Health shoulders the significant task of procuring and disseminating medicines essential for over 10,000 public health facilities [1]. The orchestration of drug distribution becomes intriguing when contemplating the journey of these vital medicines. The Central Medical Store Depot (CMSD) takes the reins, ensuring a seamless flow as it dispatches medicines directly to the Township Public Health Department (TPHD) and upper echelons. This dynamic process eliminates the need for excessive pondering on transportation logistics for TPHD and upper levels [2].

The intricate dynamics of transportation and delivery of supplied drugs at first-level health facilities in Myanmar warrant careful consideration. An overwhelming 81.97% of store sites operated outside public sector supply mechanisms, relying on motorcycles as the primary mode of transport, covering an average distance of less than 3 hours from the township drugstore. Notably, the financial responsibility for transportation costs was distributed among health facility-owned budgets (25.68%), health facility leader-owned budgets (42.68%), and drug supply management staff-owned budgets (28.96%), emphasizing the economic strain faced by these entities, as reported by 68.31% of respondents. A glaring procedural deficiency was highlighted, with 90.16% of cases lacking SOPs for drug transportation and delivery, indicating a critical gap in operational guidelines. The SWR analysis accentuated functional weaknesses in the absence of SOP guidelines and carriage plans, compounded by the dearth of government-owned vehicles. Counteractively, strengths were discerned in the meticulous observation of drug carriage pathways and thorough checks on drug items. However, the absence of a public sector transportation mechanism and the financial burden associated with transportation costs emerged as significant risks in this intricate supply chain. Strategic interventions were imperative to address these challenges and enhanced the efficiency and reliability of drug transportation and delivery systems at the first-level health facilities in Myanmar. Comprehensive research studies and interventions in comparable global contexts should be explored for potential insights and best practices to inform tailored improvements in Myanmar's supply chain.

## Waste management

The examination of waste management practices associated with supplied drugs across 183 first-level health facilities in Myanmar unveiled a nuanced scenario. Predominant techniques encompassed burial pits (49.18%), incineration (62.84%), and sharp pits (55.19%). Notably, a significant proportion maintained an adequate supply of safety boxes (78.14%) and waste bins (69.40%). Moreover, there was a commendable disposal rate for used needles, with 80.87% ensuring safe disposal, and 57.38% employing a color-coded system for waste bins. Nevertheless, challenges persisted, as 32.24% incurred costs for waste management, and nearly 39% of these spent over 5000 kyats. The financial burden was primarily shouldered by health facility-owned budgets (22.95%), health facility leader-owned budgets (25.68%), and drug supply management staff-owned budgets (48.09%).

A comprehensive SWR analysis underscored strengths in the functional areas of SOP guidelines, diverse waste management techniques, adequate provision of safety boxes, proper utilization of waste bins, and effective disposal procedures—all scoring above the 50th percentile. However, a notable weakness was evident in the availability of public services for waste management, with a score below the 50th percentile. The absence of trained waste handlers

and the financial costs associated with waste management emerge as potential risk factors, given their scores above the 50th percentile.

To enhance waste management practices, Myanmar's health facilities could benefit from fortifying public services, ensuring training for waste handlers, and exploring cost-effective waste management strategies. These findings underscored the importance of tailored interventions to address specific weaknesses and risks in waste management within the first-level health facilities of Myanmar.

In evaluating overall performance across different functional areas in our study, the waste management system emerged as the top performer, with the highest capability maturity level score. This notable achievement can be attributed to strategic provisions for waste management stemming from diverse programs within the first-level public health facilities. Take, for instance, the proactive measures implemented to tackle the COVID-19 pandemic, which equipped health facilities with essential infrastructures and tools for both disease prevention and the proper disposal of vaccines. However, our investigation unearthed a significant caveat–the burden of waste management fell directly on the shoulders of the drug management staff, demanding their direct involvement and financial commitment. The absence of trained waste handlers and the associated costs of waste management pose potential risks, casting shadows over the sustained effectiveness of this critical functional area.

## Logistics management information system

The evaluation of the LMIS in 183 public health facilities reveals a mixed landscape. Alarmingly, a substantial 62.84% lacked SOP guidelines for the operational definitions of LMIS and exclusively relied on paper-based systems. While the performance percentages for various LMIS documentation aspects, such as invoice vouchers, stock ledger books, and health facility stock report books, exhibited robust figures, there are notable exceptions like Bin cards (9.84%) and discrepancy report forms (32.24%). The consistency of drug balance between stock books and stores remained a critical concern, with an average percentage of only 13.66%.

Challenges plaguing LMIS implementation included frequent stockouts of tools, delayed feedback, difficulties in data filling, analysis, and retrieval, usage of different tool versions, reliance on outdated tools, inadequate training (62.84%), insufficient human resource capability (19.13%), and a shortage of staff (38.25%). These findings underscored the urgent need for targeted interventions to streamline and fortify LMIS processes. Benchmarking against established international best practices, alongside tailored training programs and resource augmentation, was imperative to enhance LMIS effectiveness in the dynamic landscape of public health facilities. This warranted collaborative efforts and knowledge exchange with global initiatives addressing similar LMIS challenges, ensuring a comprehensive and sustainable improvement. The distribution of SOP/Guidelines within the LMIS functional domain emerged as a weak link, with each department relying solely on a pen-and-paper-based system for LMIS operations. On a positive note, document maintenance and record-keeping exhibited robust practices across most departments. However, when delving into the challenges faced by LMIS, a deficiency in formal training and a scarcity of essential resources like stock, registers, and reporting forms came to the forefront, posing hurdles to the seamless functioning of the system.

## Quality control procedures

The assessment of quality control procedures for supplied drugs across 183 drugstore sites revealed notable gaps, with 71.04% lacking SOP guidelines. Approximately half of the sites conducted monthly checks on various parameters such as packing damages, brand integrity,

seal and instruction damages, changes in colors, sedimentation of injections, cracks, humidity, leaking, drying of oil, crushed/broken drugs, loss of drugs from blister cards, sticky drugs, unusual odors, and expiry dates. In-depth scrutiny of physical damage conditions encompassed considerations like overlay (27.32%), dusty drugs and packing (80.33%), signs of pest infestation (74.86%), signs of water damage (66.67%), presence of waste bins (69.40%), "No Smoking" signboard (42.62%), presence and condition of fire extinguishers (24.59% and 19.67% respectively), presence of sandbags near the drug store (13.11%), preventive measures for pest infestation (27.87%), and data quality assessment (29.51%). The average scores for checking storage quality conditions and physical drug damages were 46.06% and 48.20%, respectively, both falling below the 50th percentile.

These findings underscored significant deficiencies in the quality control measures implemented in drugstore sites, warranting immediate attention and improvement. A previous study [20] on pharmaceutical quality control and storage practices can provide valuable insights for enhancing these procedures. Implementing robust SOPs, investing in regular training programs, and adopting advanced technologies for monitoring drug quality can contribute to a more effective and reliable quality control framework within first-level health facilities.

## Conclusion

The study reveals that the overall supply chain maturity at first-level public health facilities is at a marginal capability level (36.35%). While some basic drug supply chain management procedures are in place, they are not consistently followed, and many systems remain manual. The findings underscore significant inconsistencies in the management functions of supplied drugs, with poor adherence to SOP guidelines. This research highlighted critical deficiencies in various aspects of the drug supply chain at first-level public health facilities in Myanmar. Gaps in training, forecasting practices, storage management, ordering and receiving processes, dispensing patterns, transportation, waste management, LMIS, and quality control procedures were identified. The findings align with international studies, emphasizing the need for standardized procedures, enhanced training, and infrastructure improvements. Urgent interventions, benchmarking against global best practices, and collaborative efforts with international organizations are recommended to address these challenges and enhance the reliability and effectiveness of the pharmaceutical supply chain in Myanmar. Future research should explore tailored strategies for improvement in specific functional areas.

## Dissemination plan

The researchers intend to disseminate the outcomes of the research to the entirety of public health practitioners within the designated research domain, the regional public health department situated in the Bago region, and the procurement and supply division operating at the Central level. Ultimately, the research findings will be showcased at the Myanmar Research Congress and subsequently published in a reputable international journal.

## Acknowledgments

Our hearts brim with gratitude towards the IR (Implementation Research) Granters from the esteemed Department of Medical Research, whose generous funding has been the lifeblood of this research endeavor. A heartfelt appreciation extends to the benevolent Deputy Director General and Deputy Directors of the Bago Regional Public Health Department, whose kind permission has paved the way for our exploration. Special thanks dance towards the Deputy Director, Assistant Directors, and the compassionate public health professionals of Pyay

District, whose invaluable in-kind support has been a beacon guiding our journey. Last but certainly not least, our sincere thanks to all the respondents, whose active participation has infused this research with vibrancy and significance.

## Author Contributions

**Conceptualization:** Thein Hlaing.

**Data curation:** Thein Hlaing, Tun Win Lat.

**Formal analysis:** Thein Hlaing, Tun Win Lat.

**Funding acquisition:** Thein Hlaing.

**Investigation:** Thein Hlaing, Tun Win Lat.

**Methodology:** Thein Hlaing.

**Project administration:** Thein Hlaing, Tun Win Lat.

**Resources:** Thein Hlaing.

**Software:** Thein Hlaing.

**Supervision:** Thein Hlaing, Tun Win Lat.

**Validation:** Thein Hlaing, Tun Win Lat.

**Visualization:** Thein Hlaing, Tun Win Lat.

**Writing – original draft:** Thein Hlaing.

**Writing – review & editing:** Thein Hlaing.

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
