## [Decision Letter · Decision Letter 0]

16 Jul 2024

PGPH-D-24-01229

Drug Supply Management at First-level Public Health Facilities: Case of Pyay District, Myanmar

Dear Dr. Hlaing,

Thank you for submitting your manuscript to PLOS Global Public Health. After careful consideration, we feel that it has merit but does not fully meet PLOS Global Public Health’s publication criteria as it currently stands. Therefore, we invite you to submit a revised version of the manuscript that addresses the points raised during the review process.

Below you will find the comments from the reviewer on your manuscript, please address these. In particular, they raise concerns on the validation of the tools used. We note that the reviewer requests that the ethical considerations are shortened, please note there is no requirement to remove the information you have included in this section but you may wish to convey the information more succinctly. 

Please note that we have only been able to secure a single reviewer to assess your manuscript. We are issuing a decision on your manuscript at this point to prevent further delays in the evaluation of your manuscript. Please be aware that the editor who handles your revised manuscript might find it necessary to invite additional reviewers to assess this work once the revised manuscript is submitted. However, we will aim to proceed on the basis of this single review if possible. 

We look forward to receiving your revised manuscript.

Kind regards,

Joanna Tindall

Staff Editor

Journal Requirements:

1. Tables should not be uploaded as individual files. Please remove these files and include the Tables in your manuscript file as editable, cell-based objects. For more information about how to format tables, see our guidelines:

https://journals.plos.org/globalpublichealth/s/tables

Additional Editor Comments (if provided):

Reviewers' comments:

Reviewer's Responses to Questions

**Comments to the Author**

1. Does this manuscript meet PLOS Global Public Health’s publication criteria? Is the manuscript technically sound, and do the data support the conclusions? The manuscript must describe methodologically and ethically rigorous research with conclusions that are appropriately drawn based on the data presented.

Reviewer #1: Yes

2. Has the statistical analysis been performed appropriately and rigorously?

Reviewer #1: Yes

3. Have the authors made all data underlying the findings in their manuscript fully available (please refer to the Data Availability Statement at the start of the manuscript PDF file)?

Reviewer #1: Yes

4. Is the manuscript presented in an intelligible fashion and written in standard English?

Reviewer #1: No

5. Review Comments to the Author

Reviewer #1: Background

1. I'd like for the authors to review all the abbreviations. Each abbreviation should be defined only once, at its first appearance in the manuscript (from introduction to conclusion). Subsequently, only the short forms should be used. 

2. The background is very vast and can be shortened.

Study settings and population 

1. The author should mention how these health facilities were selected and clearly describe the sampling method.

Preparation of data collection tools 

1. Have you validated your data collection tools?

Assessment of Capacity Maturity of drug supply chain

1. Various functional areas, such as the capability maturity level of capacity-building in drug supply management, the capability maturity level of drug forecasting and planning etc. can be expressed as outcome measures and written in the form of a table. 

Data collection

1. The time period of data collection needs to be mentioned under the heading of data collection. 

Ethical consideration

1. This paragraph needs to be shortened. There isn't a need to explain everything. Only the ethical approval number and details of the source of ethical approval are enough.

Results

Result texts need to be shortened. Information written as texts, is already there in tables therefore duplication should be avoided.

6. PLOS authors have the option to publish the peer review history of their article (what does this mean?). If published, this will include your full peer review and any attached files.

**Do you want your identity to be public for this peer review?** For information about this choice, including consent withdrawal, please see our Privacy Policy.

Reviewer #1: No

---

## [Decision Letter · Decision Letter 1]

16 Aug 2024

Drug Supply Management at First-level Public Health Facilities: Case of Pyay District, Myanmar

PGPH-D-24-01229R1

Dear Mr Hlaing,

We are pleased to inform you that your manuscript 'Drug Supply Management at First-level Public Health Facilities: Case of Pyay District, Myanmar' has been provisionally accepted for publication in PLOS Global Public Health.

Best regards,

Bashar Haruna Gulumbe

Academic Editor

Reviewer Comments (if any, and for reference):

Reviewer's Responses to Questions

**Comments to the Author**

1. If the authors have adequately addressed your comments raised in a previous round of review and you feel that this manuscript is now acceptable for publication, you may indicate that here to bypass the “Comments to the Author” section, enter your conflict of interest statement in the “Confidential to Editor” section, and submit your "Accept" recommendation.

Reviewer #1: All comments have been addressed

2. Does this manuscript meet PLOS Global Public Health’s publication criteria? Is the manuscript technically sound, and do the data support the conclusions? The manuscript must describe methodologically and ethically rigorous research with conclusions that are appropriately drawn based on the data presented.

Reviewer #1: Yes

3. Has the statistical analysis been performed appropriately and rigorously?

Reviewer #1: Yes

4. Have the authors made all data underlying the findings in their manuscript fully available (please refer to the Data Availability Statement at the start of the manuscript PDF file)?

Reviewer #1: Yes

5. Is the manuscript presented in an intelligible fashion and written in standard English?

Reviewer #1: No

6. Review Comments to the Author

Reviewer #1: (No Response)

7. PLOS authors have the option to publish the peer review history of their article (what does this mean?). If published, this will include your full peer review and any attached files.

**Do you want your identity to be public for this peer review?** For information about this choice, including consent withdrawal, please see our Privacy Policy.

Reviewer #1: No
